# The choice of the objective function in flux balance analysis is crucial for predicting replicative lifespans in yeast

**Barbara Schnitzer**[1,2], **Linnea Österberg**[1,2,3], **Marija Cvijovic**[1,2]*

1 Department of Mathematical Sciences, Chalmers University of Technology, Gothenburg, Sweden, 2 Department of Mathematical Sciences, University of Gothenburg, Gothenburg, Sweden, 3 Department of Biology and Biological Engineering, Chalmers University of Technology, Gothenburg, Sweden

* marija.cvijovic@chalmers.se

## Abstract

Flux balance analysis (FBA) is a powerful tool to study genome-scale models of the cellular metabolism, based on finding the optimal flux distributions over the network. While the objective function is crucial for the outcome, its choice, even though motivated by evolutionary arguments, has not been directly connected to related measures. Here, we used an available multi-scale mathematical model of yeast replicative ageing, integrating cellular metabolism, nutrient sensing and damage accumulation, to systematically test the effect of commonly used objective functions on features of replicative ageing in budding yeast, such as the number of cell divisions and the corresponding time between divisions. The simulations confirmed that assuming maximal growth is essential for reaching realistic lifespans. The usage of the parsimonious solution or the additional maximisation of a growth-independent energy cost can improve lifespan predictions, explained by either increased respiratory activity using resources otherwise allocated to cellular growth or by enhancing antioxidative activity, specifically in early life. Our work provides a new perspective on choosing the objective function in FBA by connecting it to replicative ageing.

## 1 Introduction

The topology of metabolic networks are well established for many types of prokaryotic and eukaryotic cells, particularly in comparison to other biochemical networks. The reconstruction of the networks is feasible due to availability of experimental data and standardised methods [1–3]. However, analysing them is more challenging, mainly because of their large size and interconnectivity [4–7]. Constraint-based mathematical modelling, such as flux balance analysis (FBA), has helped to unravel features of the metabolic reconstruction [8, 9]. The FBA approach represents chemical reactions through mass-balance and steady-state assumptions on the network components. The optimisation problem is formulated by a set of linear constraints and is classically solved for a set of target objectives. Despite generally large solution spaces, FBA models were able to accurately predict exchange fluxes, growth rates and metabolic switches in chemostat and batch cultures in various conditions and organisms [3, 10–13]. Still, predicting individual fluxes or enzymes remains a challenge [14]. An extensive

model itself from https://github.com/cvijoviclab/IntegratedModelMetabolismAgeing.

**Funding:** This work was supported by the Swedish Research Council (VR2016-03744 and VR2017-05117) to MC and the Swedish Foundation for Strategic Research (FFL15-0238) to MC.The funders had no role in study design, data collection and analysis, decision to publish, or preparation of the manuscript.

**Competing interests:** The authors have declared that no competing interests exist.

amount of work has been dedicated to making resulting optimal flux distributions more realistic, by adding biologically motivated constraints, such as resource [15], enzyme [16], proteome [17] or thermodynamic constraints [18, 19].

While the choice of the objective function also has a vast impact on the possible solutions, and ultimately determines how the fluxes can be distributed across the network, it has received less attention. Schuetz et al. investigated a set of objective functions in an *E.Coli* stoichiometric network model and fitted the simulated fluxes to C-based flux data in different conditions, concluding that maximal energy (ATP) or biomass production are most accurate to describe the data [20]. Moreover, it was concluded that the objective functions that fit the data best can be condition-dependent. More recent studies showed similar results in the budding yeast *S. cerevisiae* [21]. The minimisation of the redox potential in the cell was further mentioned as a potential objective [22]. Algorithms to infer an objective function to a FBA model using experimental flux data also suggested maximal growth as the best choice [23, 24]. Furthermore, combinations of different objectives have been investigated using multi-objective optimisation [25, 26] or yield optimisation [27, 28].

Altogether, the consensus objective doesn't exist. Further, even though motivated by evolutionary arguments, the selection criteria for objective functions in FBA in previous studies arose from flux data from the metabolism, and have not been directly coupled to evolutionary properties such as reproduction or ageing. In this work, we therefore aimed to systematically investigate the effect of the objective function in flux balance analysis on the replicative ageing in cells, using budding yeast as a model organism. While rationalising experimental data in order to decide which objective function fits the best under certain conditions is feasible, it is more challenging to experimentally measure the objective function and study its consequences on long-term dynamic effects like ageing. For that reason, we exploit our recently published multi-scale mathematical model of yeast metabolism and ageing [29]. The central carbon metabolism including the creation of reactive oxygen species (ROS) is represented by an enzyme-constrained flux balance model, that is further constrained by the regulation upon oxidative stress and nutrient availability, and connected to a dynamical model of damage accumulation and growth, including discrete cell division event. We can therefore simulate the effect of molecular changes in the metabolism on observables on the cellular level, such as the number of daughter cells produced, i.e. the replicative lifespan, and the time between cell divisions, i.e. the generation time.

In this work, we analysed the effect of commonly used objectives in FBA on evolutionary important features in yeast wildtype cells, shedding light on the old question of the choice of the objective function in FBA from a new theoretical perspective.

## 2 Materials and methods

### 2.1 Enzyme-constrained flux balance analysis

The metabolism is modelled by a flux balance analysis (FBA) model with enzymatic constraints [16]. We denote the fluxes by $\mathbf{v}$ $[mmol(gDWh)^{-1}]$, the stoichiometry by $S$ and the enzyme usages by $\mathbf{e}[mmol(gDW)^{-1}]$ and solve the linear program (LP) in (1)-(7). The total enzyme usage is restricted by the total enzyme pool $\sigma f P_{tot}$, a factor consisting of the average saturation $\sigma$, the fraction of enzymes covered in the model $f$ and the total protein content of the cell $P_{tot}[g(gDW)^{-1}]$. The optimisation problem is formulated as:

$$max./min. \qquad z_1 = \mathbf{c}^T\mathbf{v} \qquad (1)$$

$$s.t. \qquad S\,\mathbf{v} = \mathbf{0} \qquad (2)$$

$$\mathbf{v}_{min} \leq \mathbf{v} \leq \mathbf{v}_{max} \tag{3}$$

$$-\sum_j \frac{n^{ij}}{k_{cat}^{ij}} v_j + e_i = 0, \quad \forall i \tag{4}$$

$$-\sum_i MW_i e_i + e_{pool} = 0 \tag{5}$$

$$\mathbf{e}_{min} \leq \mathbf{e} \leq \mathbf{e}_{max} \tag{6}$$

$$0 \leq e_{pool} \leq \sigma f P_{tot} \tag{7}$$

Each included enzyme $i$ mediates a reaction $j$ with a rate $k_{cat}^{ij}$ and a stochiometry $n^{ij}$ (mainly relevant of enzyme complexes), and has a molecular weight $MW_i[kDa = g(mmol)^{-1}]$.

## 2.2 Multi-scale model of yeast metabolism and replicative ageing

We exploited our previously published multi-scale model (yMSA), incorporating modules for the metabolism, regulation and damage accumulation in *S. cerevisiae* yeast cells [29]. In the model, the metabolism is represented by an enzyme-constrained FBA model of the central carbon metabolism. The regulatory network consists of a vector-based Boolean representation of the Snf1, PKA, TOR, Yap1 and the Sln1 pathways. A transcriptional layer constitutes the connection between regulation and the metabolism, and effectively constrains the usage of enzymes, depending on the activity of transcription factors in the Boolean model and subsequent up- or down-regulation of enzymatic genes. The input layer of the Boolean model, in turn, is determined by the optimal fluxes through the FBA model that is optimised with a particular objective function. The optimal fluxes of the regulated FBA model are then used to feed a dynamic (ODE) model of damage accumulation and cell growth, which is solved for one time step. Over time, the fraction of functional proteins decreases due to damage accumulation processes (metabolic damage formation at rate $f_m$, non-metabolic damage formation at rate $f_0$ and damage repair $r_0$), that are partly caused by the creation of reactive oxygen and nitrogen species in the metabolism. The asymmetric distribution of protein damage at cell division displays another major cause of the damage accumulation in the model. As a consequence, the cell has a decreasing amount of functional enzymes available to maintain cellular growth and maintenance. At the same time, it is assumed that the non-growth associated maintenance cost, such as damage repair, increases the more damage the cell has. If the cell has managed to produce enough biomass, cell division occurs. The FBA model becomes infeasible when damage levels are too high, and in that case the cell is considered dead. In that way, the model allows to simulate replicative ageing as the accumulation of damage, which is steered by the metabolism and the regulatory network. All mathematical and computational details of the model as well as model parameters can be found in [29]. In particular, we used parameters of a cell with a non-metabolic damage formation $f_0 = 0.0001$ and damage repair $r_0 = 0.0005$, as well as a regulation factor 0.04. This parameter combination leads to a wildtype yeast cell with 23 divisions and an average generation time of around 1.5h, that was generated using the parsimonious maximal growth objective in the FBA model. This cell is considered the reference cell for this work, for which the effect of the objective function was investigated.

## 2.3 Two-stage approach for the optimisation

Each optimisation strategy in our model is described by two successive optimisations (lexico-graphic method) [26]. We optimise the first objective, constrain the corresponding flux (or sum of fluxes) to the optimal value allowing to violate it by some factor $\epsilon_1 \leq 1$, and then optimise the second objective.

The first optimisation corresponds to solving the LP defined in (1)-(7). The following second optimisation then becomes:

$$max./min. \qquad z_2 = \mathbf{d}^T\mathbf{v} \tag{8}$$

$$s.t. \qquad constraints \ (2) - (7) \tag{9}$$

$$\mathbf{c}^T\mathbf{v} \begin{cases} \geq z_1(1 - \epsilon_1) & if \ z_1 \ was \ maximised \\ \leq z_1(1 + \epsilon_1) & if \ z_1 \ was \ minimised \end{cases} \tag{10}$$

In exactly the same way as the constraint introduced in ((10)), we allow to violate the second optimal value $z_2$ by a factor $\epsilon_2 \leq 1$ for the following regulation step in the integrated model that imposes stricter constraints on enzymes ($\mathbf{e}_{min,max}$) depending on their regulation. In our framework, it is necessary to give a bit of flexibility to the system to reallocate the enzyme usages as a result of the gene regulation to avoid that the systems becomes infeasible.

By doing two successive optimisations, we automatically force a certain priority to the first objective, and within the resulting solution space we choose the solution that also optimises the second objective up to the defined flexibilities.

In this work, we tested several different individual objective functions as well combinations of them to investigate their effect on the replicative life of the cell: maximal growth (biomass reaction), minimal glucose uptake (glucose uptake reaction), maximal and minimal ATP production (sum of all reactions that produce ATP), minimal NADH production (sum of all reactions that produce NADH), and maximal non-growth associated maintenance (NGAM reaction). In addition, we check both the direct solution of the optimisation procedure and the parsimonious solution, i.e. the solution that also minimises the sum of all fluxes and the total enzyme usage implemented as an additional optimisation. For the latter objective, we only allowed a flexibility according to the solver precision, to find the most flux- and enzyme-efficient solution given the previous objectives.

When comparing fluxes, we calculated the average of each flux (absolute, or normalised by the glucose uptake rate) within a metabolic phase. We investigated the relative change $\Delta$ between the respective non-parsimonious $\mathbf{v}$ and the parsimonious $\mathbf{v}^p$ solutions (Fig 2, S3 and S4 Figs), calculated by

$$\Delta_i = \frac{|v_i^p - v_i|}{|v_i|}, \qquad \forall \ i. \tag{11}$$

Note, that before calculating the change, we transformed the metabolic network back to a network with reversible reactions and removed isoenzymes, to avoid double-counting of fluxes. As a consequence, fluxes can have negative values depending on the direction. Here we are only interested in the change, and not the direction, explaining the use of absolute values.

With the same logic, we compared the fluxes between only maximal growth as an objective and additional parsimony or NGAM maximisation (Fig 3).

## 2.4 Simulation details

All simulations were performed in the programming language Julia version 1.6 [30] and were run on a normal computer with 2.3 GHz Dual-Core Intel Core i5 and 8GB RAM, using the JuMP optimisation package and Gurobi as a solver for the linear programs. Relevant simulation code and the underlying data of all figures can be downloaded from https://github.com/cvijoviclab/AgeingObjectiveFunction and details about the model itself from https://github.com/cvijoviclab/IntegratedModelMetabolismAgeing.

# 3 Results

## 3.1 Maximal growth is the most realistic objective for reaching wildtype yeast lifespans

The objective of the FBA model is naturally changing the distribution of optimal fluxes in the metabolic network, which in turn influence damage accumulation and lifespan in our model. To quantify the changes in the replicative lifespan and the generation times, we simulated the lifespan of cells using different objective functions in the FBA model of the metabolism. In particular, we tested objectives comparable to previous studies [20, 21, 31]: maximal growth, minimal glucose uptake, maximal and minimal ATP production, minimal NADH production, and maximal non-growth associated maintenance (NGAM). We chose a lexicographic approach with up to two successive optimisations, denoted with 1 and 2 in the subscripts. After each optimisation we allowed a violation of the respective optimal value $z_{1,2}$ by $\epsilon_{1,2} \cdot z_{1,2}$, with $0 \leq \epsilon_{1,2} \leq 1$, in the following optimisations to ensure flexibility and feasibility. We investigated a range of $\epsilon_{1,2}$ to better understand the consequence of this parameter. We did not constrain the usage of glucose, neither from above nor from below, such that its uptake rates is purely determined by the objective and the existing reaction, enzymatic and regulatory constraints. For that reason, a minimisation step as fist objective causes simulated cells to either not or only grow by slowly taking in nutrients and not dividing. In this work we focus on replicative ageing, thus we allow nutrient-rich environment and disregard effects of nutrient limitations.

The simulations showed that the choice of the objective functions has a vast impact on the replicative lifespan and the generation times (Fig 1A and 1B, left panels). Without having maximal growth as either the first or second optimisation the cells do not reach high enough growth rates, and in many cases do not divide at all or only 1–2 times with long generation times. We therefore conclude that maximal growth is crucial for replicative ageing.

Further, there is no consistent effect on the replicative lifespans for increased flexibility in the objective values ($\epsilon_{1,2}$). However, the average generation times are increased for increased $\epsilon_1$ if maximal growth is the first objective, while it is decreased when the same objective is used as second. The flexibility $\epsilon_2$ generally seems to be less influential on the two ageing characteristics.

We defined wildtype cells as cells that divide between 20 and 30 times with average generation time of 1.5 to 2.3 hours [32–36]. In addition, wild diploid cells, are bigger and divide fast with a generation time about 45 minutes in early stage of ageing. The reported 90 min lifespan in the literature is mostly based on haploid strains. Further, the cell generation time also increase close-to-linear during the ageing process [37] of haploid strains.We observed that only few combinations of objective functions and $\epsilon_{1,2}$ generated what we denote as wildtype cells (Fig 1C, left panel). In those cases, maximal growth as an objective is always included. Additionally, maximal NGAM, as both first or second objective, seems to make flux distributions more realistic in the sense that more wildtype cells can be generated.

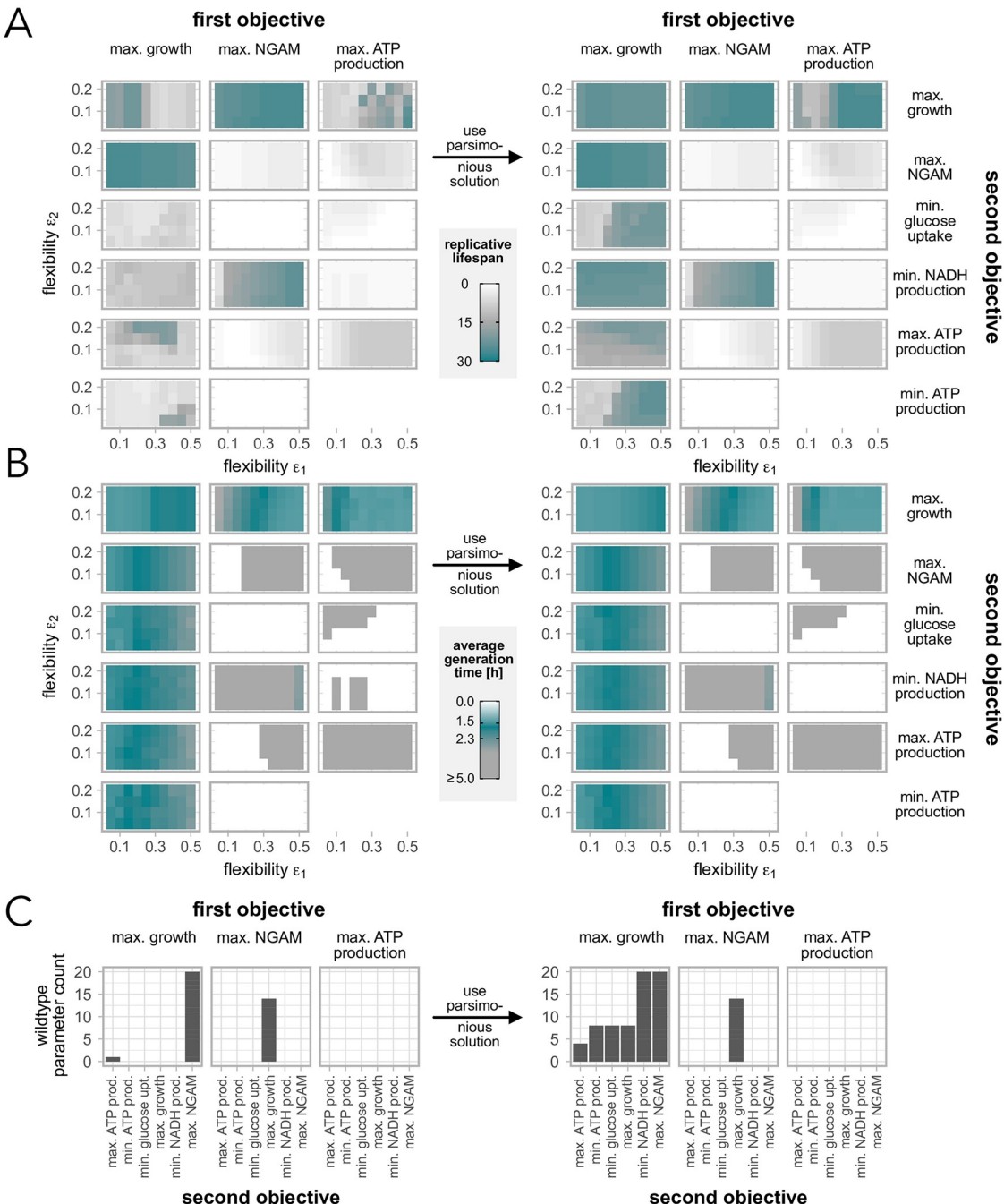

**Fig 1. Effect of objective functions on replicative lifespans and generation times.** Replicative lifespans (A) and generation times (B) for cells simulated with different objectives and flexibilities $\epsilon_{1,2}$ using the non-parsimonious (left) and the parsimonious (right) solutions. (C) Counts of how many parameter combinations lead to replicative lifespans between 20 and 30 divisions, and generation times between 1.5 and 2.3h, denoted as wildtype cells.

## 3.2 Using the parsimonious solution can cause a rearrangement of fluxes leading to increased lifespans

A common approach to decrease the possible solution space of FBA models is to take the parsimonious solution, i.e. the solution with the minimal sum of fluxes through the network [38].

In enzyme-constrained FBA the minimal enzyme usage is an optional addition. Biologically, it is justified by the assumption that cells would always choose the most efficient nutrient consumption and enzyme usage to create energy. To test the effect of this assumption on ageing characteristics, we repeated the simulation described previously, but used the parsimonious solution for each parameter set (Fig 1A and 1B, right panels). This means that after the maximally two optimisations, we performed another optimisation and minimised the sum of all fluxes and enzymes usages, given the optimal values of the previous optimisations.

The simulation showed that the average generation times remained unchanged when introducing the flux- and enzyme-efficient solution. However, the replicative lifespans could be increased when maximal growth was the first objective or when optimising for maximal ATP production and maximal growth, while all other cases were not affected. The use of the parsimonious solution had a particularly strong effect if at the same time the flexibility $\epsilon_1$ was larger than 30%. As a consequence, the number of observed wildtype cells increased substantially when primarily maximising for growth (Fig 1C, right panel, and S1 Fig). It is also worth noting that applying an additional parsimonious optimisation has a negligible effect if maximal NGAM was included.

To study how and why an efficient usage of the resources can increase the replicative lifespan, we analysed the respective changes of the mean fluxes over time, focusing on parameters in the regime where we observed the biggest deviations between the solutions: $\epsilon_1 \geq 30\%$ for maximal growth as a first objective and not NGAM as a second, as well as maximal ATP production followed by maximal growth. Since the metabolism changes significantly during ageing, we separated the lifespan of each cell in two phases according to [13, 29]. Phase I is considered the maximal growth phase dominated by a fermenting metabolism. Phase II starts when the initial growth rate drops and the metabolism makes more and more use of cellular respiration. We averaged the fluxes over the respective phases, and investigated the absolute difference between the non-parsimonious and the parsimonious solution, as well as the relative differences. In the latter case, we used fluxes that were normalised by the glucose uptake rate in each time step in the model, allowing us to better compare changes across the pathways. Hence, positive values generally correspond to an increase, and negative ones to a decrease. A flux that is switched off has a relative change of -1.

Naturally, many fluxes were reduced or remained constant by using the parsimonious solution, but interestingly, there is also a substantial percentage of fluxes in most of the included pathways that is increased, indicating a rearrangement of fluxes (Fig 2A). The metabolic phases show only minor differences. In phase II, there are more pathways with a higher fraction of fluxes that are decreased compared to phase I.

To better understand the effect of using the parsimonious solution on the system, we ordered the fluxes by the pathway they belong to. We observed that the objective function has an impact on the rearrangement of the fluxes, and pathways are affected differently dependent on its choice (Fig 2B and S2 Fig). Including a second optimisation on top of maximal growth forces adaptions in more pathways compared to only maximal growth. Prioritising maximal ATP production before maximal growth, exhibits the largest deviations. In phase I, all objectives have in common that there are noticeable relative (Fig 2B) and absolute (S2 Fig, not normalised) changes of fluxes in the exchange reactions, the oxidative phosphorylation and oxidative stress pathway, anaplerotic reations, the TCA cycle, in mitochondrial transport and other. In particular, the oxidative phosphorylation and oxidative stress pathways, anaplerotic reactions and other have at least 10% of the fluxes that are changed by more than 100%, indicating the largest rearrangements (S3 Fig). In phase II, all pathways except the galactose metabolism are affected by using the parsimonious solution. Still, fluxes connected to oxidative stress showed the largest relative changes (S3 Fig).

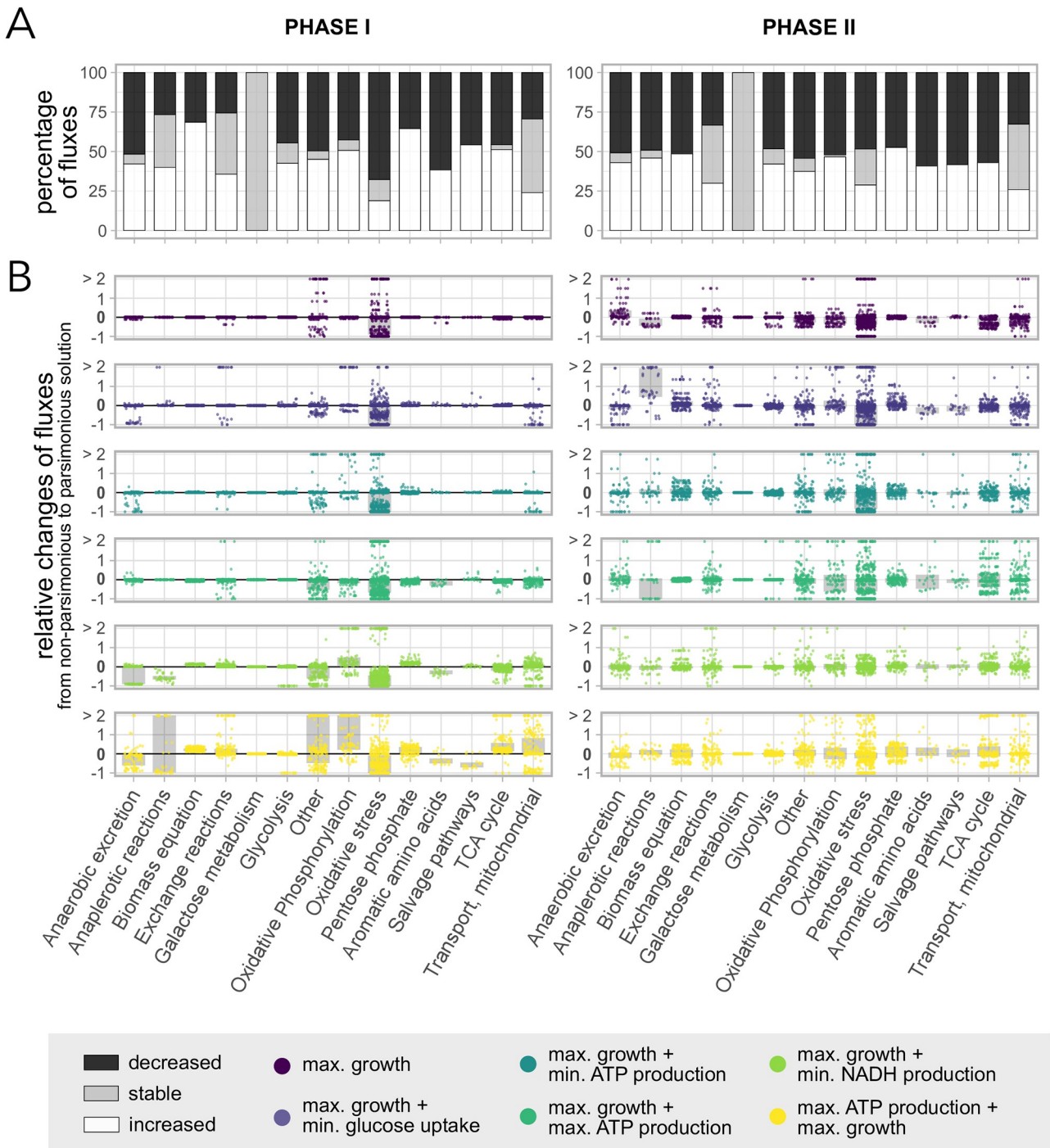

**Fig 2. Relative changes of normalised fluxes between the non-parsimonious and the parsimonious solution.** We limited the analysis to objectives that show a large increase in the replicative lifespans as a consequence of imposing parsimony. Included are 20 parameter combination with $\epsilon_1 \geq 0.3$ and $\epsilon_2 \leq 0.2$ per investigated objective (Fig 1). All fluxes are normalised by the glucose uptake rate and averaged over a metabolic phase (left: I, right: II). (A) Percentage of fluxes that are increased (white), unchanged (grey) or decreased (black) in all included pathways in the FBA model, when going from the non-parsimonious to the parsimonious solution. All objective functions are merged in this plot. (B) Relative changes of all fluxes in the respective pathways, when going from the non-parsimonious to the parsimonious solution. The grey bars indicate the interquartile ranges of the distributions.

To elucidate how the changes of the fluxes in the oxidative stress pathway correlate to an increased replicative lifespan in the model, we investigated the fluxes in that particular pathway in more detail (S4 Fig). While reactive oxidative species (ROS), such as superoxide and hydrogen peroxide (superoxide oxidoreductase), can be produced more compared to the non-parsimonious solution, antioxidants, such as hydrogen peroxide catalse (in both metabolic phases) and glutathione reductase and peroxidase (predominantly in phase II), are able to redirect the fluxes and prevent protein damage formation by neutralising ROS. Thus, using the parsimonious solution leads to lower damage production rates, presumably responsible for slower ageing and longer lifespans.

### 3.3 Maximising for a non-growth associated ATP cost has similar but stronger effects than using the parsimonious solution

The maximal non-growth associated maintenance (NGAM) is an additional reaction in the metabolic model that removes ATP from the systems, under the assumption that it is needed for non-growth related maintenance tasks. In the model, it is further assumed to increase over the replicative life of a cell. Even though it is an extra cost for the cell, we previously saw that maximising the NGAM is not affected by parsimony and can lead to realistic features of replicative ageing without using a flux-efficient solution. To understand how this specific objective affects the fluxes through the metabolic network, we studied the relative alterations of the fluxes, normalised by the glucose uptake rate and averaged over the respective metabolic phase, through all included pathways in two cases: (1) solely maximal growth compared to maximal growth using the parsimonious solution, and (2) solely maximal growth compared to maximal growth and maximal NGAM as a second objective. Similar to before, we focused on parameter sets with $\epsilon_1 \geq 30\%$.

Even though the effect of additionally maximising NGAM on characteristics of replicative ageing is similar to using the parsimonious solution, we found that the respective flux distributions differ significantly from each other (Fig 3A). Maximising NGAM comes with a larger rewiring of the fluxes across the network, that in most pathways corresponds to a positive relative change compared to only maximal growth as an objective. Phase I is furthermore more affected than phase II. Interestingly, in phase I glycolysis is decreased but the biomass production per glucose increased. At the same time, absolute biomass production (not normalised by glucose uptake) is inhibited since the generation times are increased for many parameter combinations (Fig 3B). This indicated that cells in phase I, which is typically dominated by fermentation, also make more use of respiration to gain energy, since it yields more ATP given the same nutrient uptake than fermentation, as well as related pathways, such as oxidative phosphorylation, TCA cycle and oxidative stress, are enhanced.

Generally, using NGAM as a second objective leads to a more flux-efficient solution in most cases (Fig 3C), however not a more enzyme-efficient (Fig 3D) solution. Again the differences are more prominent in phase I.

### 3.4 Denoted wildtype cells mainly differ in the times they spend in the respective metabolic phases

We investigated in which way cells with distinct objectives differ from each other, by selecting the parameter combinations that led to realistic wildtype cells. We compared measures, such as the time the cells spend in each of the two metabolic phases, how many times they divided within this time and how much protein damage they produced.

While all wildtype cells by definition have similar replicative lifespans and generation times, we demonstrated that the choice of the objective function affects how long cells can remain in

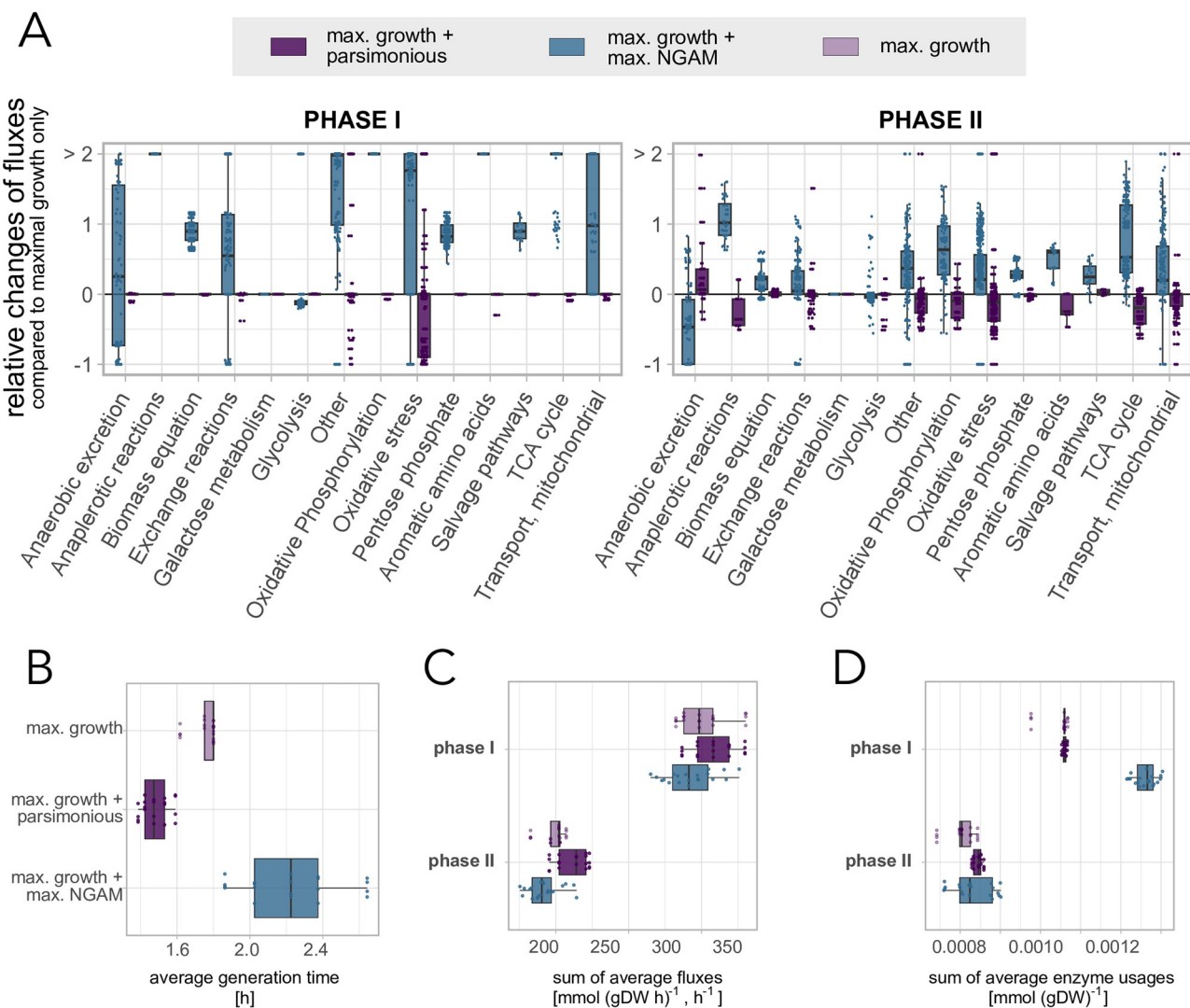

**Fig 3. Comparison between using the parsimonious solution or an additional optimisation of NGAM.** We compared only maximal growth, parsimonious maximal growth, and maximal growth plus maximal NGAM. Included are 20 parameter combination with $\epsilon_1 \geq 0.3$ and $\epsilon_2 \leq 0.2$ per investigated objective (Fig 1). All fluxes are normalised by the glucose uptake rate and averaged over a metabolic phase (left: I, right: II). (A) Relative changes of all fluxes in the respective pathways for parsimonious maximal growth and maximal growth plus maximal NGAM, both in relation to maximal growth only. (B) Average generation times for each included parameter set, sorted and coloured by the respective objective function. (C) Sum of all fluxes for each included parameter set, coloured by the respective objective function. (D) Sum of all enzyme usages, that were averaged over the respective metabolic phase, for each included parameter set, coloured by the respective objective function.

the maximal growth phase I, before entering phase II, as much as the number of divisions in the respective phases (Fig 4). Generally, the longer the cells spend in a phase, the more often they divided and the more damage they accumulated during that phase. Cells optimised with maximal growth in combination with maximal NGAM are an exception and have comparable damage levels at the end of phase I, even though they divided more often in that phase than cells with other objectives.

Thus, there are several strategies to reach wildtype characteristics, which are mainly coupled to how the metabolic network is exploited.

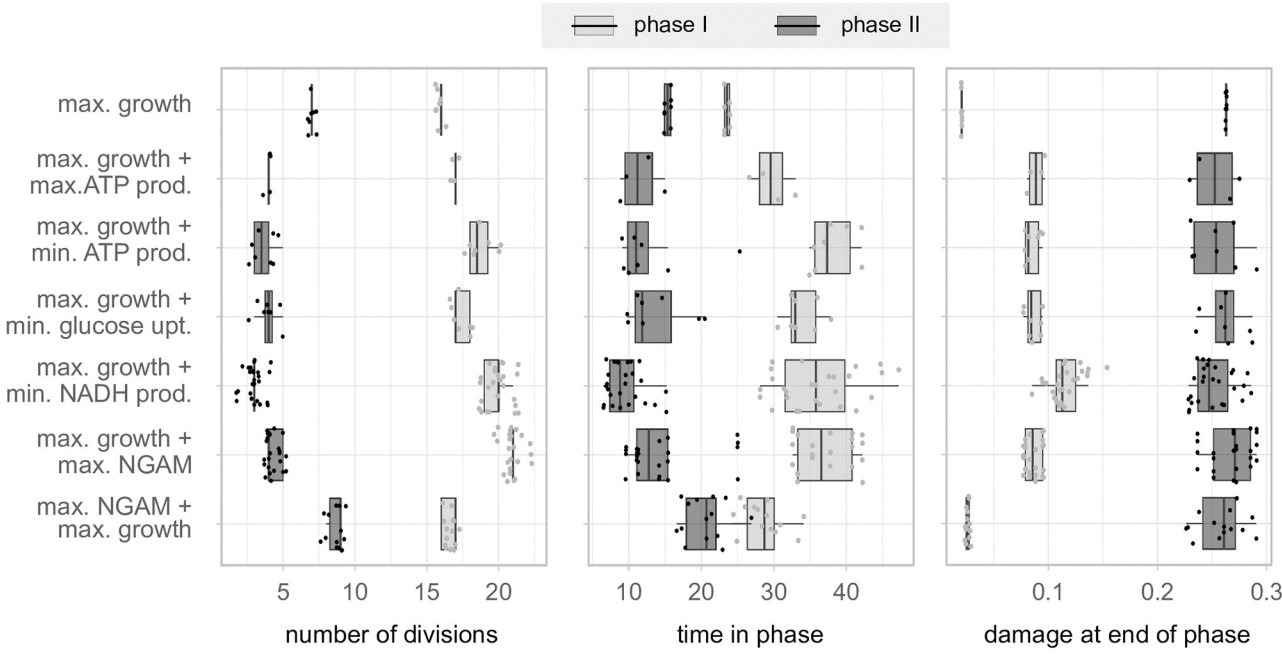

**Fig 4. Comparison of overall properties of wildtype cells in the metabolic phases.** Number of divisions and time spent in each metabolic phase, and the damage at the end of each phase for all parameter combinations that lead to wildtype cells (replicative lifespans between 20 and 30 divisions, and generation times between 1.5 and 2.3h, see S1 Fig). The damage at the end of phase II corresponds to the fraction of damaged proteins when the cell dies. The cells are grouped by the respective combination of objective functions. Here, we only present the parsimonious cases.

## 4 Discussion

In FBA modelling, the objective function is closely related to and thus often motivated by arguments from evolutionary biology. In evolutionary biology, fitness is generally composed of viability, mating success and fertility [39], hence, the replicative lifespan. However, the generation time is also an important feature during competitive growth. Here, we presented a systematic analysis of objective functions in the context of replicative ageing, utilising an enzyme-constrained FBA model of the central carbon metabolism of budding yeast cells, embedded in a published integrated model of nutrient signalling, metabolism and protein damage accumulation [29]. We found that maximal growth is the most important objective with regard to the replicative lifespan, in line with previous studies [20, 21, 23]. We further showed that an additional optimisation can improve the predictions of features of replicative ageing. We focused particularly on maximal growth as a first objective, followed by either the usage of the parsimonious solution or an additional maximisation of the non-growth associated maintenance (NGAM).

In the simulations, we applied a lexicographic procedure [26], consisting of typically two successive optimisations, with the first having a higher priority than the second. On top of that, we distinguished between the usage of the direct outcome of the optimisation algorithm and the parsimonious solution [38]. This approach is similar but not identical to traditional multi-objective optimisation, that can, for example, be solved by optimising a weighted sum of the individual objectives [25, 26], or to yield optimisation [27, 28], where instead a fraction of objectives is optimised, often used in industrial applications. In the context of ageing, we chose to apply several successive optimisations, which allowed us a simple analysis and biological

interpretation of the parameters and results. Solely the parsimonious assumption alone reflects a classical multi-objective optimisation using a sum as an objective. Regardless of the set or sole objective function used, they are based on strong assumptions. In addition, constraint-based models generally suffer from uncertainties in the underlying experimental data, which can lead to over-constraining the model. To address this limitation, we introduced the flexibilities $\epsilon_{1,2}$ to the model, that can conceptually be compared to attributing weights to the objectives. Without the flexibility, no objective achieved a wildtype behaviour, reflecting the strong assumptions made.

Utilising the model, we could test hypothesis on the objectives and their effects on the metabolism and ageing. We found that a major difference between cells without and with parsimony is the increased usage of antioxidants to prevent protein damage production, specifically important in the first metabolic phase. Besides the toxic effect of ROS as precursors to protein damage, low levels of ROS have been shown to be beneficial for the robustness of cells independent of the metabolic phase, for instance by acting as signalling molecules [40–43]. The second metabolic phase is accompanied by small changes in both directions in almost all pathways, likely a consequence of different preconditions when cells exit the first and enter the second phase. In addition, in the latter phase, cells generally have a decreased functional enzyme pool due to the advanced progression of ageing, which automatically forces the cell to be more efficient in their usage, thus more parsimonious. Parsimony can also be based on evolutionary arguments, however it is hard to interpret the results. The corresponding objective comprises the sum of all individual fluxes and enzymes, that are generally not equally important across the network but have equal weight in the optimisation.

Using the more interpretable second objective of maximal non-growth associated maintenance (NGAM) has similar effects as parsimony on the replicative lifespans, even without imposing efficiency on the system. NGAM increases the ATP demand, which results in enhanced respiratory activity, inhibited cellular growth and prolonged generation times. Respiration has a higher ATP yield per glucose which can generally explain increased fluxes through respiratory pathways. Since the enzyme pool is limited, and respiration is less enzyme efficient, the cell likely has to simultaneously decrease the growth rate. This gives the cell more time to repair damage. Thus, ageing is not accelerated by the rearrangement of fluxes, being crucial for reaching high replicative lifespans.

Taken together, both extensions to maximal growth discussed above can have a beneficial affect on the lifespan, by rearranging fluxes across the network. Simultaneously, we can interpret those objectives as a trade-off between growth or reproduction and maintenance, in line with the disposable soma theory of ageing that states that ageing is the consequence of this trade-off [44, 45]. Here, we could confirm that there is a balance between the two objectives, and pushing growth to the absolute limit can be disadvantageous for individual cells. Giving more priority to maintenance could actually improve average growth over the lifespan. Increased respiration and prolonged generation times also prevented protein damage production, and therefore only had minor effects on the lifespans, emphasising the importance of this balance.

In summary, there are innumerable different flux distributions that result in a specific cellular growth rate or other experimentally testable output, both in FBA and likely also in reality. Here, we showed that when working with cells under evolutionary pressure, maximal growth as an objective is inevitable. Adding more objectives, such as parsimony or a maximal non-growth related maintenance cost can be helpful to pick more biologically reasonable flux distributions. We demonstrated that robustness in lifespans can be achieved by a combination of balance and flexibility in allocating the resources. Hence, we provided a new perspective on

the choice of the objective function from a theoretical point of view, putting the objective in FBA in the context of evolutionary variables such as reproduction and replicative ageing.

## Supporting information

**S1 Fig. Parameter combinations that are considered yeast wildtype cells.** All parameters $\epsilon_{1,2}$ marked in black generate wildtype cells with a replicative lifespans between 20 and 30 divisions, and generation times between 1.5 and 2.3h, in our model. Based on Fig 1.
(PDF)

**S2 Fig. Absolute changes of fluxes between the non-parsimonious and the parsimonious solution.** We limited the analysis to objectives that show a large increase in the replicative lifespans as a consequence of imposing parsimony. Included are 20 parameter combination with $\epsilon_1 \geq 0.3$ and $\epsilon_2 \leq 0.2$ per investigated objective (Fig 1). We averaged the fluxes over the two metabolic phases (left: I, right: II). The results are similar to Fig 2, but here each average flux is neither scaled by the glucose uptake rate, nor by the respective non-parsimonious flux, but is an absolute difference.
(PDF)

**S3 Fig. Fluxes with a large relative change between the non-parsimonious and the parsimonious solution.** Percentage of fluxes in the respective pathways with relative change of at least 100% when imposing parsimony, being a subset of the fluxes shown in Fig 2B. All objective functions are merged in this plot.
(PDF)

**S4 Fig. Effects of parsimony on the oxidative stress pathway.** Relative changes of fluxes between the non-parsimonious and the parsimonious solution. Included are 20 parameter combination with $\epsilon_1 \geq 0.3$ and $\epsilon_2 \leq 0.2$ per investigated objective (Fig 1). We limited the analysis to objectives that show a large increase in the replicative lifespans as a consequence of imposing parsimony. Each flux is normalised by the glucose uptake rate and averaged over the metabolic phase (left: I, right: II).
(PDF)

## Acknowledgments

We would like to thank members of the Cvijovic lab for valuable input.

## Author Contributions

**Conceptualization:** Barbara Schnitzer, Linnea Österberg, Marija Cvijovic.

**Formal analysis:** Linnea Österberg.

**Investigation:** Barbara Schnitzer, Linnea Österberg.

**Methodology:** Barbara Schnitzer.

**Project administration:** Marija Cvijovic.

**Supervision:** Marija Cvijovic.

**Visualization:** Barbara Schnitzer.

**Writing – original draft:** Barbara Schnitzer, Linnea Österberg.

**Writing – review & editing:** Barbara Schnitzer, Marija Cvijovic.

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
