## [Decision Letter · Decision Letter 0]

27 Jun 2022

PONE-D-22-14577The choice of the objective function in flux balance analysis is crucial for predicting replicative lifespans in yeastPLOS ONE

Dear Dr. Cvijovic,

Thank you for submitting your manuscript to PLOS ONE. After careful consideration, we feel that it has merit but does not fully meet PLOS ONE’s publication criteria as it currently stands. Therefore, we invite you to submit a revised version of the manuscript that addresses the points raised during the review process.

We look forward to receiving your revised manuscript.

Kind regards,

Hong Qin

Academic Editor

PLOS ONE

Journal Requirements:

2. Please upload a copy of Figures 1 to 3 , to which you refer in your text. If the figure is no longer to be included as part of the submission please remove all reference to it within the text.

Additional Editor Comments:

One major finding in the current version is that assuming maximal growth is essential for reaching realistic lifespans. Replicative lifespan have substantial natural variations, up to 200%. Growth and replicative lifespan are also influenced by growth media. So, we hope that this main conclusion is re-examined in the context of different genotypes and different growth conditions.

Reviewers' comments:

Reviewer's Responses to Questions

**Comments to the Author**

1. Is the manuscript technically sound, and do the data support the conclusions?

Reviewer #1: Partly

2. Has the statistical analysis been performed appropriately and rigorously? 

Reviewer #1: Yes

3. Have the authors made all data underlying the findings in their manuscript fully available?

Reviewer #1: Yes

4. Is the manuscript presented in an intelligible fashion and written in standard English?

Reviewer #1: Yes

5. Review Comments to the Author

Reviewer #1: In this work, Schnitzer et al aimed to investigate the relation between function in flux balance analysis on replicative lifespan (RLS) model of yeast aging. They developed a multi-scale model by incorporating a boolean connection to gene networks based on transcriptome and metabolome to analyze how commonly used objectives in FBA affected evolutionarily important cellular features in yeast cells to identify factors with significant impact on lifespan. Although there is some overlap with the previously published manuscript (https://doi.org/10.1101/2022.03.07.483339) from the same research group, this is an interesting study and would be of interest to the field.

Overall manuscript is well written. However, I have one major concern that needed to be addressed:

The study is based on simulation and their model could well predict the experimental observation for average RLS and generation times of WT yeast cells. This is a very important observation and needed to be further verified for other strains. There are publicly available resources of transcriptome, proteome and metabolome data for different yeast knock-out strains or interventions (e.g. TOR1 or caloric restriction ). The model should be further tested whether it also predicts features of replicative aging in these strains/conditions along with RLS values.

Minor concerns:

There should be more details for prediction of lifespan and growth time and prediction of increased lifespan. Since these are the major findings of the paper, it should be comprehensively described in method and discussed throughout the discussion section.

As far as I know, there is no single model that can predict yeast RLS with this much precision. So that this study will be in the high interest of the community. However,without testing/verifying their model in other strains/conditions, the validity and the significance of the findings would not be impactful.

6. PLOS authors have the option to publish the peer review history of their article (what does this mean?). If published, this will include your full peer review and any attached files.

Reviewer #1: No

---

## [Author Response · Author response to Decision Letter 0]

2 Aug 2022

Additional Editor Comments:

One major finding in the current version is that assuming maximal growth is essential for reaching realistic lifespans. Replicative lifespan have substantial natural variations, up to 200%. Growth and replicative lifespan are also influenced by growth media. So, we hope that this main conclusion is re-examined in the context of different genotypes and different growth conditions.

We would like to thank the editor for taking the time to review our paper and for providing a careful and thorough evaluation of the manuscript. 

This work is based on our multi-scale model of yest replicative ageing that has now been accepted for publication in PLoS Computational Biology (https://doi.org/10.1371/journal.pcbi.1010261). In that paper we observe the variation of replicative life span between 17 – 32 divisions. We are aware of natural variation of RLS, like those reported in Kaya et al 2021 (https://doi.org/10.7554/eLife.64860) or Janssens and Veenhoff 2016 (https://doi.org/10.1371/journal.pone.0167394). However, when it comes to ageing studies in yeast, standard laboratory strains show the RLS of typically 20-25 divisions, and this is the standard metrics used and we used this as a guidance for our studies. Further, in the work presented in this manuscript we use the same growth media that is a standard for FBA modeling in general. 

We agree with the comment that many features are influenced by growth media, however, we aimed here to focus on standard media and examine typical objectives used in FBA community to make our work as applicable as possible for genome-scale modelers. Exploring those in the wider context is interesting, but this was not the aim of our study. 

Reviewers' comments:

Reviewer's Responses to Questions

Comments to the Author

1. Is the manuscript technically sound, and do the data support the conclusions?

Reviewer #1: Partly

2. Has the statistical analysis been performed appropriately and rigorously?

Reviewer #1: Yes

3. Have the authors made all data underlying the findings in their manuscript fully available?

Reviewer #1: Yes

4. Is the manuscript presented in an intelligible fashion and written in standard English?

Reviewer #1: Yes

5. Review Comments to the Author

Reviewer #1: In this work, Schnitzer et al aimed to investigate the relation between function in flux balance analysis on replicative lifespan (RLS) model of yeast aging. They developed a multi-scale model by incorporating a boolean connection to gene networks based on transcriptome and metabolome to analyze how commonly used objectives in FBA affected evolutionarily important cellular features in yeast cells to identify factors with significant impact on lifespan. Although there is some overlap with the previously published manuscript (https://doi.org/10.1101/2022.03.07.483339) from the same research group, this is an interesting study and would be of interest to the field.

Overall manuscript is well written. However, I have one major concern that needed to be addressed:

The study is based on simulation and their model could well predict the experimental observation for average RLS and generation times of WT yeast cells. This is a very important observation and needed to be further verified for other strains. There are publicly available resources of transcriptome, proteome and metabolome data for different yeast knock-out strains or interventions (e.g. TOR1 or caloric restriction ). The model should be further tested whether it also predicts features of replicative aging in these strains/conditions along with RLS values.

We would like to thank the reviewer for taking the time to review our paper and for providing a careful and thorough evaluation of the manuscript. 

The work presented here is based on our multi-scale model of yeast replicative ageing, which has been recently accepted for publication in PLoS Computational Biology (https://doi.org/10.1371/journal.pcbi.1010261). In this paper we have verified several of the model prediction (like deletion and overexpression, including the Snf1, Tor1 and Tor2 and many others) with publicly available data. Further, we were able to simulate transition of different metabolic phases and make predictions of enzyme perturbations that affect RLS. We do agree that model needs further testing, but that is outside the scope both of the published model and the work presented in this manuscript. 

Minor concerns:

There should be more details for prediction of lifespan and growth time and prediction of increased lifespan. Since these are the major findings of the paper, it should be comprehensively described in method and discussed throughout the discussion section.

As far as I know, there is no single model that can predict yeast RLS with this much precision. So that this study will be in the high interest of the community. However, without testing/verifying their model in other strains/conditions, the validity and the significance of the findings would not be impactful.

Thanks for noticing the prediction power of the model when it comes to RLS. We believe that this clearly shows that to study ageing, holistic models are needed to be able to understand the synergistic effect of multiple processes affecting ageing. Previous ageing models focused only on individual processes, and this can be a reason why they lack the predictive power of our published model. In the paper where we introduced the multiscale model, we provide full details (model description, simulation details, predictions and validations). Similar to the above comment, we are aware that more validations are needed and is something our group is working with. 

The aim of this paper submitted to PloS One and which is reviewed here is solely form the theoretical point of view on the role of the objective function, as common factor when it some to FBA models, but we investigate this in light of replicative lifespan. Using the correct objective in FBA to engineer “long-living” yeast could improve industrial titres and manufacturing of cell-based diagnostic devices. 

We provide here the abbreviated summary of the model and full details of the optimization approach relevant for this study, together with the extensive simulation instructions and details on github. 

6. PLOS authors have the option to publish the peer review history of their article (what does this mean?). If published, this will include your full peer review and any attached files.

---

## [Decision Letter · Decision Letter 1]

29 Aug 2022

PONE-D-22-14577R1The choice of the objective function in flux balance analysis is crucial for predicting replicative lifespans in yeastPLOS ONE

Dear Dr. Cvijovic,

Thank you for submitting your manuscript to PLOS ONE. After careful consideration, we feel that it has merit but does not fully meet PLOS ONE’s publication criteria as it currently stands. Therefore, we invite you to submit a revised version of the manuscript that addresses the points raised during the review process.

We look forward to receiving your revised manuscript.

Kind regards,

Hong Qin

Academic Editor

PLOS ONE

Journal Requirements:

Additional Editor Comments:

I suggest the author to make minor revisions based on the following two references.

The first study on the natural variation of yeast replicative lifespan was published in 2006, by Hong Qin, Meng Lu, "Natural variation in replicative and chronological life spans of Saccharomyces cerevisiae"

Experimental Gerontology, Volume 41, Issue 4,Pages 448-456.

The cell generation time (cell cycle) also increase close-to-linear during the aging process, as reported by Li et al, 2020 "A programmable fate decision landscape underlies single-cell aging in yeast", Science, v369, pp325-329. The Li 2020 paper is based on haploid cell.

In addition, wild isolate are diploid cells, are bigger and divide fast with a generation time about 45 minutes in early stage of aging. The reported 90 min lifespan in the literature are mostly based on haploid strains.

Reviewers' comments:

Reviewer's Responses to Questions

**Comments to the Author**

1. If the authors have adequately addressed your comments raised in a previous round of review and you feel that this manuscript is now acceptable for publication, you may indicate that here to bypass the “Comments to the Author” section, enter your conflict of interest statement in the “Confidential to Editor” section, and submit your "Accept" recommendation.

Reviewer #1: All comments have been addressed

Reviewer #2: (No Response)

2. Is the manuscript technically sound, and do the data support the conclusions?

Reviewer #1: Yes

Reviewer #2: Partly

3. Has the statistical analysis been performed appropriately and rigorously? 

Reviewer #1: Yes

Reviewer #2: Yes

4. Have the authors made all data underlying the findings in their manuscript fully available?

Reviewer #1: Yes

Reviewer #2: Yes

5. Is the manuscript presented in an intelligible fashion and written in standard English?

Reviewer #1: Yes

Reviewer #2: Yes

6. Review Comments to the Author

Reviewer #1: The authors have satisfactorily addressed most of my concerns. Current version of the manuscript is suitable for publication in PLos One.

Reviewer #2: (No Response)

7. PLOS authors have the option to publish the peer review history of their article (what does this mean?). If published, this will include your full peer review and any attached files.

Reviewer #1: No

Reviewer #2: No

---

## [Author Response · Author response to Decision Letter 1]

13 Sep 2022

The first study on the natural variation of yeast replicative lifespan was published in 2006, by Hong Qin, Meng Lu, "Natural variation in replicative and chronological life spans of Saccharomyces cerevisiae"

Experimental Gerontology, Volume 41, Issue 4,Pages 448-456.

The cell generation time (cell cycle) also increase close-to-linear during the aging process, as reported by Li et al, 2020 "A programmable fate decision landscape underlies single-cell aging in yeast", Science, v369, pp325-329. The Li 2020 paper is based on haploid cell.

In addition, wild isolate are diploid cells, are bigger and divide fast with a generation time about 45 minutes in early stage of aging. The reported 90 min lifespan in the literature are mostly based on haploid strains.

We would like to thank Editor for point out to those references. We have now incorporated them into the ms. 

Below are the answers to the Reviewers comments attached as pdf. These are the comments from our original submission to PLoS Comp Bio and were already addressed when we transferred the ms to PLOS ONE. 

Reviewer #1: In this work, Schnitzer et al aimed to investigate the effect of the objective function in flux balance analysis (FBA) on yeast replicative aging. They analyzed how commonly used objectives in FBA affected evolutionarily important features in yeast cells. 

The authors showed that the type of the objective function has a big say on yeast replicative lifespan as well as generation times, concluding that maximal growth was essential for replicative aging. Also, their findings indicated that using the parsimonious solution could lead to increased lifespan through rearrangement of fluxes.

Their model could reproduce close-to-experimental values for average replicative lifespan and generation times of wild-type yeast cells. The model also predicts features of replicative aging with distinct metabolic phases. 

This is an interesting piece of work as it brings a novel way of thinking on the choice of the objective function by putting the objective in FBA in the context of evolutionary variables. The finding that maximal growth is the most important objective in regard to replicative lifespan will have significant impact. 

We would like to thank the reviewer for a very careful and thorough evaluation of the manuscript. We have received very much appreciated, constructive and valuable comments and suggestions. We carefully addressed all comments which lead to a revision of the manuscript and the supplementary information.

I would ask the authors to revise their manuscript along the following two points:

1. Regarding their underlying multi-scale model, it would be good to explain/inform the reader about which parameters have been empirically known and which have not. Also, any quantitative assessment of the robustness of the overall model should ideally be made. 

We provided a thorough description and motivation of all parameters in the linked paper that introduces the model (https://doi.org/10.1101/2022.03.07.483339). While the focus of this work is not the model, but how we used it to study the effect of the objective function on evolutionary observables such as lifespan, we here refer the reader to that paper instead, where all computational and mathematical details can be found, but also emphasised again the specific parameters that correspond to the reference cell (lines 280-285, and section Simulation Details). 

2. There are previously-published yeast-aging-focused or time-dynamic single-cell-modeling-focused papers that should have been cited due to their relevance to the different sections of this manuscript: 

http://doi.org/10.1016/j.celrep.2019.07.082

http://doi.org/10.1186/s12859-019-2921-3

http://doi.org/10.1186/s12918-015-0240-5

We disagree with the reviewer and do not find the publications relevant for this particular work concerning the objective function of the FBA model, while above mentioned paper are centered around dynamic models. However, we acknowledge the relevance of those papers for the model itself and therefore included them in the linked publication (https://doi.org/10.1101/2022.03.07.483339).

---

## [Decision Letter · Decision Letter 2]

27 Sep 2022

PONE-D-22-14577R2The choice of the objective function in flux balance analysis is crucial for predicting replicative lifespans in yeastPLOS ONE

Dear Dr. Cvijovic,

Thank you for submitting your manuscript to PLOS ONE. After careful consideration, we feel that it has merit but does not fully meet PLOS ONE’s publication criteria as it currently stands. Therefore, we invite you to submit a revised version of the manuscript that addresses the points raised during the review process.

We look forward to receiving your revised manuscript.

Kind regards,

Hong Qin

Academic Editor

PLOS ONE

Journal Requirements:

Additional Editor Comments (if provided):

Hello Mariga,

After communication with a reviewer, I would suggest you to briefly mention the following reference in your revised manuscript.

Song, R., Acar, M. Stochastic modeling of aging cells reveals how damage accumulation, repair, and cell-division asymmetry affect clonal senescence and population fitness. BMC Bioinformatics 20, 391 (2019). https://doi.org/10.1186/s12859-019-2921-3

I agree with the reviewer that Song19 is a relevant reference to your manuscript.

Hong

Reviewers' comments:

Reviewer's Responses to Questions

**Comments to the Author**

1. If the authors have adequately addressed your comments raised in a previous round of review and you feel that this manuscript is now acceptable for publication, you may indicate that here to bypass the “Comments to the Author” section, enter your conflict of interest statement in the “Confidential to Editor” section, and submit your "Accept" recommendation.

Reviewer #2: (No Response)

2. Is the manuscript technically sound, and do the data support the conclusions?

Reviewer #2: Yes

3. Has the statistical analysis been performed appropriately and rigorously? 

Reviewer #2: Yes

4. Have the authors made all data underlying the findings in their manuscript fully available?

Reviewer #2: (No Response)

5. Is the manuscript presented in an intelligible fashion and written in standard English?

Reviewer #2: Yes

6. Review Comments to the Author

Reviewer #2: (No Response)

7. PLOS authors have the option to publish the peer review history of their article (what does this mean?). If published, this will include your full peer review and any attached files.

Reviewer #2: No

---

## [Author Response · Author response to Decision Letter 2]

27 Sep 2022

After communication with a reviewer, I would suggest you to briefly mention the following reference in your revised manuscript.

Song, R., Acar, M. Stochastic modeling of aging cells reveals how damage accumulation, repair, and cell-division asymmetry affect clonal senescence and population fitness. BMC Bioinformatics 20, 391 (2019). https://doi.org/10.1186/s12859-019-2921-3

I agree with the reviewer that Song19 is a relevant reference to your manuscript.

 We have now incorporated this reference into the ms.

---

## [Editor Report · Decision Letter 3]

29 Sep 2022

The choice of the objective function in flux balance analysis is crucial for predicting replicative lifespans in yeast

PONE-D-22-14577R3

Dear Dr. Cvijovic,

We’re pleased to inform you that your manuscript has been judged scientifically suitable for publication and will be formally accepted for publication once it meets all outstanding technical requirements.

Kind regards,

Hong Qin

Academic Editor

PLOS ONE

Additional Editor Comments (optional):

Hi Marija,

Thanks for your patience with working with me and the reviewers. I acknowledge that the minor revision is completed, and I think there is no more issues with the reviewers.

Hong Qin
---

## [Editor Report · Acceptance letter]

3 Oct 2022

PONE-D-22-14577R3 

The choice of the objective function in flux balance analysis is crucial for predicting replicative lifespans in yeast 

Dear Dr. Cvijovic:

I'm pleased to inform you that your manuscript has been deemed suitable for publication in PLOS ONE. Congratulations! Your manuscript is now with our production department. 

Kind regards, 

on behalf of

Dr. Hong Qin 

Academic Editor

PLOS ONE